# Factors in Immigrant Children's Use of Physician and Dentist Visits, Hospital Care, and Prescribed Medication in the United States

Tyrone C. Cheng [1,*] and Celia C. Lo [2]

1    School of Social Work, University of Alabama, Little Hall, Tuscaloosa, AL 35401, USA
2    Peraton, Defense Personnel and Security Research Center, Seaside, CA 93955, USA; celiaclo@yahoo.com
*    Correspondence: tyronecheng@yahoo.com or ccheng@retiree.ua.edu

**Abstract:** Applying the behavioral model of health services utilization to data from the United States, this study examined immigrant children's use of physician and dentist visits, hospital care, and prescribed medication. We employed data describing 9759 immigrant parents and children who participated in 2021's National Survey of Children's Health. Logistic regression results negatively linked physician visits to child health, child age, Asian children, fathers, lower parent education, lower family income, uninsured children, and parent's U.S. residence under 5 years. Dentist visits were positively associated with child age, girls, Hispanic children, parent education, family income, public/private health insurance coverage, and U.S.-born children, but such use was associated negatively with Asian children. Hospital use was positively associated with poor child health, Black children, children of "other" race/ethnicity, younger parent age, enrolled in health insurance, and parent's U.S. residence under 5 years. The use of prescribed medication was negatively associated with Asian children, younger child age, lower parent education, uninsured child, and lack of English proficiency. The paper's conclusion suggests policymakers expand Medicaid and CHIP eligibility among immigrant children and suggests community education to foster awareness of children's physical and oral health needs and of Medicaid, CHIP, and prescription assistance programs. The conclusion calls for healthcare providers and social workers to accommodate and respect immigrants' traditional health-related beliefs, showing cultural competence.

**Keywords:** physician visits; dentist visits; hospital care; prescribed medication; immigrant children

## 1. Introduction

Seeking to develop guidance for an expansion of publicly funded health services in the United States, our study explored factors in immigrant children's use of U.S. physician and dentist visits, hospital care, and prescribed medication. Prior studies in the United States indicate that 53.1% to 85.1% of immigrant children surveyed had visited a physician [1–5], while 39.8% to 68.5% had visited a dentist [3,6–8]. Additionally, one study reported 14% to 20% of infants in immigrant families had been hospitalized within the past year [9]. Studies have found immigrant children visit physicians 4% to 16% less often than non-immigrant children do [2,3,7] and visit dentists 17.6% less often [4]. However, studies also show in the U.S., that immigrant children are more likely than non-immigrant children to have used prescribed medication [10].

*Literature Review*

To describe access to health care among immigrant children in the United States, our study applied the behavioral model of health services utilization [11–13]. Our application focused specifically on the children's use of physician and dentist visits, hospital care, and prescribed medication. In Aday and Andersen's model, these four services are linked to factors including perceived need (i.e., health status), sociodemographic characteristics

(e.g., gender, age), social structural factors (e.g., race/ethnicity, parent education), financial resources (e.g., family income, health insurance), community resources (e.g., proximity to health services), and relevant government policies (e.g., Medicaid, Children's Health Insurance Program [CHIP]).

Unsurprisingly, the medical needs of immigrant children in the U.S. are related to their use of health services. A negative association has been reported between immigrant children's health and their likelihood of visiting a physician or a dentist [5,14]. Although physical health and oral health are not generally considered interchangeable, published studies suggest that oral health and access to dentist visits do play a role in overall health [15,16]. One earlier study in the U.S. found 42% of immigrant children to have poor oral health [6]; another found children's receipt of dental services to be more likely when parents believed it is important to maintain oral health [17]. Additionally, studies link immigrant children's use of hospital care negatively to their physical health [18]. Such published findings indicate that immigrant children exhibiting poor physical or oral health need to seek health services.

The research literature from the U.S. presents mixed results overall on potential links between immigrant children's sociodemographic characteristics and their use of health services. For instance, it was reported that younger immigrant girls were more likely than older ones to visit physicians [5] and dentists [14], but also that older immigrant children were more likely than younger ones to visit dentists [6]. Furthermore, *no* link appeared between immigrant children's age (or gender) and their use of physician or dentist visits [1,19]. One published study found no association between immigrant children's gender and their likelihood of using U.S. hospital care [19]. Another, in contrast, reported that in the U.S., older immigrant children and immigrant girls were relatively likely to use prescribed medication, compared to younger immigrant children and immigrant boys [10].

Additionally, immigrant children's use of health services in the United States appears to be associated with certain social structural factors. But findings from this area of research have also been mixed. For instance, while some studies found immigrant children of minority race/ethnicity to be relatively unlikely, compared to White immigrant children, to visit physicians [4,5,10] or dentists [4,14], researchers have, at least once, found Hispanic immigrant children to be more likely to use dental services than White immigrant children were [8]. And some studies observed *no* association between race/ethnicity and children's taking of prescribed medication [10] or visiting physicians, dentists, and hospitals in the U.S. [19]. The literature from the U.S. additionally suggests that immigrant children with relatively well-educated parents are comparatively likely to use health services versus immigrant children with less-educated parents. Studies report an association in positive direction between parent education and immigrant children's likelihood of physician visits [1,5,10] and dentist visits [14]. On the other hand, at least one study from the U.S. reported parent education to have no association with immigrant children's physician visits, dentist visits, or hospitalizations [19]. Finally, parent employment status, with its involvement in families' health insurance coverage, probably affects immigrant children's access to health services. Yet one U.S. study did find that immigrant children whose parents were employed visited dentists less often than their counterparts with unemployed parents [14].

Within the literature on immigrant children in the U.S., mixed results also characterize studies of a possible relationship between family financial resources and the children's use of health services. Some studies show children from higher-income families to be more likely than children from lower-income families to visit physicians [1,5,10] and dentists [14,17]. One reviewed study reported immigrant children in lower-income families to be relatively likely to use prescribed medication [10], while another observed no link between family income and immigrant children's use of physician, dentist, or hospital visits [19].

Research in the U.S. to date strongly links immigrant parents' possession of family health insurance coverage to their children's health services use. When immigrant

parents are enrolled for such coverage, their children are more likely to visit physicians [1,3,5,10,14,19,20], dentists [3,14,19], and hospitals [19], and are also more likely to use prescribed medication [10] compared to children of immigrant parents without family health insurance. According to the U.S. literature, immigrant children are more likely than non-immigrant children to participate in Medicaid, CHIP, and similar public insurance programs designated for low-income families [3]. Published findings on how such participation may shape the use of health services are inconsistent. For example, one prior study reported that immigrant children enrolled in Medicaid were relatively likely to visit dentists [8], yet another study observed no link between immigrant children's Medicaid enrollment and their physician visits [14]. Furthermore, low-income immigrant children's use of prescribed medication is probably fostered by dedicated prescription assistance programs available in the U.S. [21]. Note that in one study, insurance coverage for medications health professionals prescribed was found to be in place for more than 40% of children in the general population having Medicaid or CHIP [22].

The research literature for the U.S. establishes that immigrant children born outside the country are comparatively unlikely to consult physicians or dentists or to visit hospitals versus U.S.-born children [3,4,14,19,23]. Our application of Andersen and Aday's model was enhanced by an accompanying investigation of acculturation as a potential factor in immigrant children's health services use. Acculturation is a process of psychological and behavioral change that results from long-term interaction with a culture beyond the culture in which one was born into [24]. How completely an individual becomes acculturated is indicated by, among other things, the proficiency demonstrated with the dominant language [25,26]. Children in U.S. immigrant families often become fluent in English quickly. Their parents, however, may not, and the parents' language struggles can hinder what they learn and understand about health services options in the U.S. [27]. This, in turn, probably hinders the use of various kinds of health services by these parents' children.

Acculturation levels among immigrants can also be measured by the length of U.S. residence. Immigrants who have resided relatively longer in the U.S. may have come to understand its medical system relatively better. Indeed, at least one study has linked, in a positive direction, immigrant parents' length of residence in the U.S. with their children's likelihood of using dentist visits [6]. In contrast, however, another study observed no association between parents' length of residence and children's use of hospital care [18]. Nothing in the reviewed literature addressed possible associations between immigrant parents' length of U.S. residence and their children's use of physician visits or prescribed medication. However, the length of parents' residence in the U.S. may affect children's health insurance coverage, which in turn appears to affect the use of health services. That is because, under the Patient Protection and Affordable Care Act, documented immigrants who have lived in the U.S. for fewer than five years cannot enroll for Medicaid (and all undocumented immigrants are excluded from Medicaid and CHIP alike) [28]. Unless documented immigrant parents have lived in the U.S. for at least five years, having a low income does not qualify their families to obtain health services via Medicaid. Low incomes tend to make private insurance unaffordable, meaning children in immigrant families new to the U.S. remain uninsured and must seek services via public clinics or emergency rooms [28].

Overall, as we have said, the reviewed literature reflects mixed results—and some meaningful gaps, as well—in the work to identify social factors possibly associated with immigrant children's use of U.S. health services. In this study, we sought to address the gaps by examining immigrant children's use of four health services: physician visits, dentist visits, hospital care, and prescribed medication use. We hypothesized that immigrant children in the U.S. would use—or not—these services in association with 11 factors that variously described them. Specifically, we posited that children's use of physician visits, dentist visits, hospital care, and prescribed medication would be (1) associated negatively with child health, child age, child minority ethnicity, and parent employment and (2) associated positively with female children, U.S.-born children, U.S.-born parents, parent

education, parent income, parent health insurance, parent U.S. residence length, and spoken English proficiency.

## 2. Materials and Methods

### 2.1. Sample

This research examined public-use data describing a sample of 9759 children extracted from the 2021 National Survey of Children's Health (NSCH). NSCH participants included 50,892 children and their caregivers in the U.S. Via self-reports, the participants provided information about their health/mental health status, access to health services, insurance coverage, and immigrant status [29]. The present sample included only families in which at least one child or parent (either biological parent or step-parent) had been born outside the U.S.

### 2.2. Measures

The present study's four dichotomous outcome variables represented the use of four health services: *physician visits*, *dentist visits*, *hospital care*, and *prescribed medication*. The physician visit (office or virtual), dentist office visit, and hospital care variables were measures of service use during the 12 months preceding an NSCH interview; the use of medication measured a child's use of a prescribed drug at the time of an NSCH interview.

The variable *child health* represented a child's need for medical care, as conceptualized by the behavioral model of health services use. *Child health* was scored by parents using a 5-point scale of health status: 5 (*excellent*), 4 (*very good*), 3 (*good*), 2 (*fair*), or 1 (*poor*). Higher scores suggested a diminishing need for medical care. *Child teeth* denoted the health of a child's teeth and mouth (i.e., oral health), measured with a similar 5-point scale: 5 (*excellent*), 4 (*very good*), 3 (*good*), 2 (*fair*), or 1 (*poor*). *Child health* served as an explanatory variable in the measuring of *physician visits*, *hospital care*, and *prescribed medication*; *child teeth* served as an explanatory variable only in the measuring of *dentist visits*. Five further explanatory variables described the sample's sociodemographic characteristics: *girl* (versus *boy*), *child age* (in years), *mother* (versus *father*), *parent age* (in years), and *married parent* (yes/no). The measures for parent characteristics provided controls during the modeling.

A second group of variables characterized children's and parents' social structural factors. A child's racial/ethnic background was indicated with 5 dummy variables: *White* (the reference), *Black*, *Hispanic*, *Asian*, and *other race/ethnicity*. The measure of *parent education* indicated the most advanced level of study a parent had completed, using offered numeric responses as follows: 1 (*8th grade or below*), 2 (*9th–12th grade*), 3 (*graduated high school or GED*), 4 (*vocational school*), 5 (*some college*), 6 (*associate degree*), 7 (*undergraduate degree*), 8 (*master's degree*), 9 (*doctoral or professional degree*). *Employed parent* indicated whether a parent reported being employed in 50 of the 52 weeks preceding the NSCH interview.

A third group of explanatory variables described family financial resources. *Family income-to-poverty ratio* indicated the percentage of federal poverty level represented by each income figure provided in the original NSCH data set. *Public health insurance* (yes/no) denoted whether a child was enrolled in Medicaid, Medical Assistance, or other public program providing health insurance. In turn, *private health insurance* (yes/no) denoted whether a child was insured by the parent's employer-sponsored health plan or other policy purchased privately by the parent. *Other health insurance* (yes/no) denoted whether a child was insured by other health insurance; the reference category was *uninsured*.

Four of our study's explanatory variables measured acculturation. The dichotomous *child born in United States* and *parent born in United States* indicated individuals' non-immigrant status. By virtue of not having immigrated to the U.S., such individuals typically do not exhibit the acculturation process characterizing most individuals who come to the U.S. from another nation. Furthermore, the dichotomous *parent's U.S. residence less than 5 years* established whether, at the time of the NSCH interview, fewer than 60 months had elapsed since an immigrant parent had reached the U.S. The measure implied eligibility and ineligibility for public medical insurance programs. Finally, the

dichotomous variable *speak English at home* indicated whether a surveyed family used mainly English when communicating orally with each other in the home. This measure was considered to indicate a family's spoken English proficiency.

### 2.3. Data Analysis

This study employed binary outcome variables, and we applied STATA logistic regression to perform estimations with robust standard errors. We used sampling weights from NSCH. Our preliminary assessment of multicollinearity problems indicated that the variables *public health insurance* and *private health insurance* generated a low tolerance statistic (<0.4) and a strong correlation ($r = -0.68$). Because it was crucial to this research to examine families' and children's health insurance status, we nevertheless retained both of these variables during final data analyses. The tolerance statistics of the other explanatory variables were 0.50 or higher. The correlations among all explanatory variables yielded by our final models were $-0.68 \leq r \leq 0.57$.

## 3. Results

### 3.1. Descriptive Statistics

Of the 9759 immigrant children in the present U.S. sample, 80.6% had visited a physician, 73.4% had visited a dentist, 2.5% had used hospital care, and 12.2% had used prescribed medication (see Table 1). On average, in the sample, children's health measured 4.5 (*very good*), and the condition of their teeth measured 4.2 (*very good*). Girls constituted 47.7% of the sample, and their mothers constituted 58.8% of the sample. On average, in the sample, the child's age was 8.3 years, and the parent's age was 41.6 years. In addition, 89.3% of parents in the sample were married. Of the sampled children, 28.1% were White, 7.2% were Black, 30.4% were Hispanic, 23.4% were Asian, and 10.9% were other race/ethnicity. Of the parents, 77.1% were employed. On average, parent education measured 6.1 (*associate degree*), and the family income-to-poverty ratio was 269.5%. Public health insurance covered 28.1% of sampled children, while private health insurance covered 64.5%, and other health insurance covered 4.5%. However, 2.9% of the children had no health insurance of any kind. Concerning birthplace, 87.9% of children in the sample and 27.0% of their parents had been born in the U.S. Of parents who had immigrated to the U.S., 4.5% had not yet been in the new country for five years. In the homes of 64.2% of the sample, the language primarily spoken was English.

**Table 1.** Descriptive statistics (n = 9759).

| Variables | | % | Mean | Range | sd |
|---|---|---|---|---|---|
| Outcome Variables | | | | | |
| Physician visits | (yes) | 80.6 | | | |
| | (no) | 19.4 | | | |
| Dentist visits | (yes) | 73.4 | | | |
| | (no) | 26.6 | | | |
| Hospital care | (yes) | 2.5 | | | |
| | (no) | 97.5 | | | |
| Prescription medication | (yes) | 12.2 | | | |
| | (no) | 87.8 | | | |
| Explanatory Variables | | | | | |
| Child health | | | 4.5 | 1–5 | 0.7 |
| Child teeth | | | 4.2 | 1–5 | 0.9 |
| Girl | | 47.7 | | | |
| Boy | | 52.3 | | | |
| Child age (year) | | | 8.3 | 0–17 | 5.2 |
| Mother | | 58.8 | | | |
| Father | | 41.2 | | | |
| Parent age (year) | | | 41.6 | 18–75 | 8.2 |
| Married parent | (yes) | 89.3 | | | |

**Table 1.** *Cont.*

| Variables | | % | Mean | Range | sd |
|---|---|---|---|---|---|
| White | | 28.1 | | | |
| Black | | 7.2 | | | |
| Hispanic | | 30.4 | | | |
| Asian | | 23.4 | | | |
| Other ethnicity/race | | 10.9 | | | |
| Parental education level | | | 6.1 | 1–9 | 2.2 |
| Employed parent | (yes) | 77.1 | | | |
| | (no) | 22.9 | | | |
| Family income-to-poverty ratio (%) | | | 269.5 | 50–400 | 130.6 |
| Public health insurance | | 28.1 | | | |
| Private health insurance | | 64.5 | | | |
| Uninsured | | 4.5 | | | |
| Child born in United States | (yes) | 87.9 | | | |
| | (no) | 12.1 | | | |
| Parent born in United States (yes) | | 27.0 | | | |
| | (no) | 73.0 | | | |
| Parent resided in U.S. less than 5 years | (yes) | 4.5 | | | |
| | (no) | 95.5 | | | |
| Speak English at home | (yes) | 64.2 | | | |
| | (no) | 35.8 | | | |

Notes: sd = standard deviation.

### 3.2. Multivariate Analysis Results

Multivariate analytical results for the four outcome variables confirmed the hypothesized models to differ, statistically, from the null models (*Wald's* $\chi^2$ = 145.79 to 308.65, $p < 0.01$; see Table 2). The likelihood of using physician visits was associated negatively with *child health* (OR = 0.69, $p < 0.01$), with *child age* (OR = 0.92, $p < 0.01$), with *Asian* (OR = 0.63, $p < 0.01$), and with *parent's U.S. residence less than 5 years* (OR = 0.63, $p < 0.05$). *Physician visits'* likelihood was also associated positively with *mother* (OR = 1.45, $p < 0.01$), with *parent education* (OR = 1.16, $p < 0.01$), with *family income-to-poverty ratio* (OR = 1.00, $p < 0.01$), and with coverage by *public health insurance* (OR = 2.56, $p < 0.01$), by *private health insurance* (OR = 2.51, $p < 0.01$), and by *other health insurance* (OR = 2.35, $p < 0.01$). *Dentist visits'* likelihood, in turn, was associated positively with *child age* (OR = 1.21, $p < 0.01$), with *girl* (OR = 1.19, $p < 0.05$), with *Hispanic* (OR = 1.43, $p < 0.05$), with *parent education* (OR = 1.12, $p < 0.01$), with *family income-to-poverty ratio* (OR = 1.00, $p < 0.05$), with coverage by *public insurance* (OR = 1.81, $p < 0.01$) or *private insurance* (OR = 1.55, $p < 0.05$), and with *child born in United States* (OR = 1.38, $p < 0.05$). *Dentist visits'* likelihood was also associated negatively with *Asian* (OR = 0.75, $p < 0.05$).

As for children's use of hospital care, its likelihood was associated negatively with *child health* (OR = 0.38, $p < 0.01$) and *parent age* (OR = 0.96, $p < 0.05$). In addition, using hospital care was associated positively with *Black* (OR = 2.23, $p < 0.05$), with *other race/ethnicity* (OR = 2.36, $p < 0.01$), with *public health insurance* (OR = 4.51, $p < 0.01$), with *private health insurance* (OR = 3.80, $p < 0.01$), with *other health insurance* (OR = 4.88, $p < 0.01$), and with *parent's U.S. residence less than 5 years* (OR = 2.43, $p < 0.05$). Finally, we observed the likelihood of using prescribed medication to be negatively associated with *child health* (OR = 0.32, $p < 0.01$) and with *Asian* (OR = 0.62, $p < 0.05$). Moreover, likelihood of using prescriptions was associated positively with *child age* (OR = 1.10, $p < 0.01$), with *parent education* (OR = 1.14, $p < 0.01$), with *public health insurance* (OR = 2.63, $p < 0.01$), with *private health insurance* (OR = 2.21, $p < 0.01$), with *other health insurance* (OR = 4.69, $p < 0.01$), and with *speak English at home* (OR = 1.72, $p < 0.01$).

**Table 2.** Logistic regressions on access to physician visits, dentist visits, hospital care, and medication prescription (n = 9759).

| Variables | Physician Visits | | Dentist Visits | | Hospital Care | | Prescribed Medication | |
|---|---|---|---|---|---|---|---|---|
| | OR | RSE | OR | RSE | OR | RSE | OR | RSE |
| Child health | 0.69 ** | 0.05 | NA | 0.11 | 0.38 ** | 0.05 | 0.32 ** | 0.03 |
| Child teeth | NA | NA | 0.92 | 0.06 | NA | NA | NA | NA |
| Child age | 0.92 ** | 0.01 | 1.21 ** | 0.02 | 1.01 | 0.03 | 1.10 ** | 0.02 |
| Girl (boy) | 0.91 | 0.10 | 1.19 * | 0.12 | 1.30 | 0.31 | 0.97 | 0.13 |
| Mother (father) | 1.45 ** | 0.18 | 1.19 | 0.14 | 1.32 | 0.35 | 1.16 | 0.18 |
| Parent age | 0.99 | 0.01 | 0.99 | 0.01 | 0.96 * | 0.02 | 0.99 | 0.01 |
| Married parent (no) | 0.90 | 0.15 | 0.99 | 0.20 | 0.70 | 0.24 | 0.95 | 0.20 |
| Black (White) | 1.10 | 0.23 | 0.88 | 0.17 | 2.23 * | 1.01 | 1.14 | 0.33 |
| Hispanic (White) | 0.92 | 0.10 | 1.42 * | 0.23 | 1.60 | 0.63 | 0.95 | 0.18 |
| Asian (White) | 0.63 ** | 0.09 | 0.75 * | 0.11 | 0.58 | 0.19 | 0.62 * | 0.14 |
| Other race/ethnicity (White) | 0.86 | 0.18 | 1.06 | 0.17 | 2.36 ** | 0.85 | 1.08 | 0.21 |
| Parent education | 1.16 ** | 0.03 | 1.12 ** | 0.04 | 1.06 | 0.07 | 1.14 ** | 0.05 |
| Employed parent (no) | 1.07 | 0.16 | 0.89 | 0.12 | 0.95 | 0.29 | 0.90 | 0.16 |
| Family income-to-poverty ratio | 1.00 ** | 0.00 | 1.00 * | 0.00 | 1.00 | 0.00 | 1.00 | 0.00 |
| Public health insurance (uninsured) | 2.56 ** | 0.58 | 1.81 ** | 0.37 | 4.51 ** | 1.53 | 2.63 ** | 0.62 |
| Private health insurance (uninsured) | 2.51 ** | 0.58 | 1.55 * | 0.33 | 3.80 ** | 1.70 | 2.21 ** | 0.54 |
| Other health insurance (uninsured) | 2.35 ** | 0.79 | 1.55 | 0.50 | 4.88 ** | 2.84 | 4.69 ** | 1.89 |
| Child born in U.S. (no) | 0.99 | 0.15 | 1.38 * | 0.26 | 0.60 | 0.21 | 0.70 | 0.19 |
| Parent born in U.S. (no) | 1.25 | 0.18 | 1.00 | 0.13 | 1.22 | 0.35 | 1.12 | 0.17 |
| Parent's U.S. residence less than 5 years | 0.63 * | 0.17 | 0.79 | 0.19 | 2.43 * | 1.17 | 1.11 | 0.47 |
| Speak English at home | 1.24 | 0.21 | 1.10 | 0.14 | 1.12 | 0.31 | 1.72 ** | 0.35 |
| Wald's $\chi^2$ = | 247.66 ** | | 250.42** | | 145.79 ** | | 308.65 ** | |

Notes: ** $p < 0.01$; * $p < 0.05$; OR = odds-ratios; RSE = robust standard errors; reference groups are in parentheses.

## 4. Discussion

Our study found that 80.6% of the sampled immigrant children had visited a physician (in an office or via telemedicine) within a year of the NSCH interview; this percentage falls within the range (53.1–85.1%) reported by earlier national studies in the United States [2,4,5]. Our study of U.S. data also found that 73.4% of the sampled immigrant children had visited a dentist within a year of the NSCH interview; that percentage is much higher than the 39.8% observed in a 2003 national study with data from a single ethnic group of immigrants [4]. Our study also found that just 2.5% of the sampled children who were of various ages had used hospital care in the year before the NSCH interview. The percentage is much lower than the 14.0% to 20.0% of immigrant infants who had, according to a prior published analysis, experienced hospitalization [9]. Finally, 12.2% of our child sample in the U.S. reported taking prescribed medication. It is suggested by these findings that many immigrant children in the U.S. visited physicians or dentists but did not use prescribed medication. An explanation could be many such children's probable use of traditional herbal or other home remedies instead of prescriptions from doctors [30].

Our study findings partially support our first hypothesis that children's use of physician and dentist visits, hospital care, and prescribed medication would be associated negatively with child health, child age, child minority ethnicity, and parent employment. Like some earlier studies, our study found links between sampled children's poor health and their likelihood of using physician visits [5,14] and hospital care [18]. Furthermore, we observed the expected negative relationship between child health and the likelihood of using prescriptions. These two findings imply that when parents perceive children's health as poor, they are likely to provide them with physician visits, hospital care, and prescribed medication. Contradicting at least one earlier study, our study observed no significant relationship between the condition of children's teeth (or their oral health) and

their likelihood of using dentist visits [17]. Such a result implies that immigrant children visit their dentists for regular check-ups.

Our study results indicate that younger immigrant children were relatively likely to visit physicians, and older ones were relatively likely to visit dentists and use prescribed medication. These findings imply that immigrant parents attend to their young children's health, while immigrant teenagers will act to maintain their own oral health and are willing to take medication prescribed by American doctors. This supports prior results [5,6,10]; so does our finding that *child age* was not associated significantly with the surveyed children's use of hospital care [19]. In other words, immigrant children would experience hospitalization when seriously sick or injured, regardless of their age.

In line with one prior published finding [8], Hispanic immigrant children in our sample were likely to use dentist visits. Close examination of the data revealed that the interaction term between Hispanic race/ethnicity and child dental condition (OR = 1.25, $p < 0.05$) was significantly associated (in the positive direction) with the likelihood of dentist visits. In other words, Hispanic immigrant children with poor oral health would not seek dental care.

The sampled Asian immigrant children, in turn, were consistently unlikely to use physician or dentist visits or take prescribed medication. One plausible explanation for such findings is Asian immigrants' ethnicity-specific cultural beliefs, which often value traditional medicines and other traditional medical treatments over what the American medical system offers [30]. On the other hand (and contradicting a prior study [19]), Black and other children of minority ethnicity in our sample of immigrants were likely to use hospital care. This implies that they elected to use hospital care in place of physician visits. In addition, our results suggest no association between parent employment and immigrant children's use of any of the four health services represented in our study's four outcome variables.

Our findings also partially support our second hypothesis that immigrant children's use of the four health services would be associated positively with a female child, U.S.-born child, U.S.-born parent, parent education, parent income, parent health insurance, parent U.S. residence length, and spoken English proficiency. Our study found no link between immigrant children's gender and their likelihood of using physician visits, hospital care, and prescribed medication; this is consistent with some prior findings [1,19]. However, contrary to some prior results describing a sample of Hispanic children [19], our study did find an association in a positive direction between *girl* and *dentist visits*. The association indicates that the surveyed immigrant girls were as likely as the immigrant boys to use *physician visits*, *hospital care*, and *prescribed medication*, although not as likely to use *dentist visits*. We also observed in our study a link in a positive direction between relatively more-educated immigrant parents and a relatively higher likelihood of their children visiting physicians and dentists and taking prescribed medication. This confirms some earlier published results [1,5,10,14], as does the lack, in our study, of any association between *parent education* and *hospital care* [19]. An implication of such findings is that relatively more-educated parents who have immigrated to the U.S. possess relatively strong knowledge or understanding of the health services available to their children via the American medical system.

Our study additionally showed that children in higher-income immigrant families were relatively likely to use physician and dentist visits alike, supporting prior findings [1,5,10,14,17]. We observed no links, however, between the *family income-to-poverty ratio* and either *hospital care* or *prescribed medication*. This near absence of associations appears consistent with prior research [19]. Also in line with the published literature is our finding, for our child sample, of the association between *public* or *private health insurance* and immigrant children's greater likelihood of using all four health services [1,3,5,10,14,19,20]. (The sole exception was the lack of association, in our study, between *other health insurance* and dentist visits.) The results of our study, then, imply that higher-income immigrant

families can afford family health insurance. Additionally, they imply that *public health insurance* facilitates low-income immigrant children's use of various health services.

Our analysis showed no link between using any of the four health services and the variable *parent born in U.S.* In addition, disaffirming certain earlier results [3,4,14,19,23], we found no associations between *child born in U.S.* and either *physician visits*, *hospital care*, or *prescribed medication*. This lack of associations implies that a non-U.S. country of origin (of parent and/or child) generally did not affect the use of health services by children in our sample. No links appeared in our study, furthermore, between *dentist visits* or *prescribed medication* and *parent's U.S. residence less than 5 years*. We did observe, however, that the children of parents arriving in the United States within five years of the NSCH interview were comparatively likely to use hospital care but comparatively unlikely to use physician visits. That is to say, more recently arrived immigrants are likely to elect hospital care for children rather than physician care. Also, our findings showed that the likelihood of an immigrant child in the sample having visited a dentist was lower when the child's birth took place outside the U.S.

Our explanatory variable, *speak English at home*, showed an association with just one of the four outcomes, prescribed medication; the association was observed in a positive direction. This finding, together with the other results, suggests that acculturation's direct impact on how immigrant children use the four health services is not pronounced. Still, any incomplete acculturation among the children's parents—notably in terms of understanding English—may indeed substantially limit parental familiarity with American health services, *indirectly* curbing the family's use of physician visits and prescribed medication. Acculturation that is incomplete suggests a parent has arrived in the U.S. relatively recently.

## 5. Conclusions

Applying Aday and Andersen's behavioral model of health services, our analysis pinpointed factors in immigrant children's use of physician visits, dentist visits, hospital care, and prescribed medication. Our study's findings have implications for the policy and practices of various health services agencies in the United States. An important one is that efforts to expand Medicaid and CHIP eligibility to include all documented immigrants, whatever their length of U.S. residence, are vital if we are to ensure access to healthcare among immigrant children. Moreover, public health agencies' community-based education programs should include classes emphasizing (respectfully) a community's cultural practices and beliefs [31]. Classes should also address immigrant parents' awareness of the oral health needs of children and of the roles of physician visits and prescribed medication in the physical health of children—in particular of children more recently arrived in the U.S. and especially if their families' ethnicity is Asian. In addition, public health education programs need to vigorously advocate for and promote immigrant families' enrollment in Medicaid, CHIP, and prescription assistance programs. This is absolutely key for those families whose parents lack much in the way of education or earning power.

It must be remembered that gaining immigrant communities' trust and building alliances within them requires public health advocates, social workers, and health services providers to adequately understand each community's traditional health-related beliefs and authentically respect them. This will include accommodating non-Western home remedies endorsed by immigrant parents along with the health practices typical of the U.S. medical system [31,32]. When immigrant parents gain trust in health services providers, misunderstandings of American health care can be dispelled, which should increase the parents' use of all four health services for their children. Trust is often built via such basic steps as providing professional interpreters and matching the ethnicity of the provider and patient.

A limitation of our study was the data set's inclusion of only a single proxy measure for the acculturation of the sampled immigrant families. This limitation demands a certain caution in generalizing our results. A further limitation inherent in our analysis stems from our data's cross-sectional nature. Because our sample comprised many parents who had

resided in the U.S. for at least five years, the sample's average parent education and family income measures were relatively high. Future research might usefully involve samples of immigrant families having less education and income and/or samples comprising parents and children alike born outside the U.S. Future research might also productively involve longitudinal data that include standardized measures of acculturation.

**Author Contributions:** The two authors made equal contributions to the preparation of the manuscript as well as to data analysis. Both authors have approved the submitted manuscript and agreed to be personally accountable for their contributions and the accuracy and integrity of their work. All authors have read and agreed to the published version of the manuscript.

**Funding:** This research received no external funding.

**Institutional Review Board Statement:** Not applicable. The institutional review board of the author's university of affiliation is exempted from reviewing the present secondary analysis of public-use data.

**Informed Consent Statement:** Not applicable. This research employed a public-use data set that does not contain identifiable information about participants. Prior to collecting the data set, the original researchers obtained the participants' informed consent.

**Data Availability Statement:** NSCH can be accessed through https://www.census.gov/programs-surveys/nsch/data/datasets.html.

**Conflicts of Interest:** The authors declare no conflict of interest.

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
