# Peer review of "Factors in Immigrant Children’s Use of Physician and Dentist Visits, Hospital Care, and Prescribed Medication in the United States"

_ejihpe, doi:10.3390/ejihpe13100159_

Round 1

Reviewer 1 Report

Good paper which adds to the growing literature on this topic.  Please see attached document.  I apologize if I couldn't offer any methodological or strong scholarly expertise in terms of peer reviewer suggestions. 

Needs subtle revision.

Author Response

  1. Revisions are highlighted in YELLOW.

  1. Our study was retitled “Factors in immigrant children’s use of physician and dentist visits, hospital care, and prescribed medication in the United States,” clarifying its inclusion solely of data obtained in the United States. Moreover, the revised manuscript’s abstract, introduction, and literature review and its sections treating the sample, discussion, and conclusions repeatedly use the expressions “in the United States” and “in the U.S.” [Reviewer 2, comment #1]

  1. We worked across the manuscript to improve writing and presentation, including the use of the expression “poor oral health” rather than “bad teeth” (see, for example, page 4, at the top). [Reviewer 1, comment #1; reviewer 2, comment #8]

  1. The revised abstract presents, as conclusions of the study, that promoting Medicaid, CHIP, and prescription-assistance programs should be beneficial, as should health services’ accommodation of cultural beliefs held by immigrants in the U.S. [Reviewer 2, comment #7]

  1. The revised introduction is 40 words shorter than the original, at 131 words rather than 171. Still, it manages to note for the first time that one reason we pursued this study was to “develop guidance for an expansion of publicly funded health services.” [Reviewer 2, comments #2 & #4]

  1. Originally, the study’s introduction and its literature review referred to three studies with Canadian samples. Mention of those three was eliminated from the revision; all other studies cited in the manuscript involved research samples comprising only residents of the U.S. [Reviewer 2, comment #4]

  1. Our revised literature review noticeably emphasizes the significance of the cited studies’ earlier findings. Three examples appear on page 4. First, text at the end of the page’s first paragraph states that “Such published findings indicate that immigrant children exhibiting poor physical or oral health need to seek health services.” Second, text beginning the second paragraph states that “The research literature from the U.S. presents mixed results, overall, on potential links between immigrant children’s sociodemographic characteristics and their use of health services.” Third, near the beginning of page 4’s third paragraph, we newly state that “But findings from this area of research have also been mixed.”

    Moving on to page 5, text starting the second paragraph states “Within the literature on immigrant children in the U.S., mixed results also characterize studies of possible relationship between family financial resources and the children’s use of health services” On page 7, at the first paragraph’s end, the revised manuscript states (supported by citation) that “Low incomes tend to make private insurance unaffordable, meaning children in immigrant families new to the U.S. remain uninsured and must seek services through public clinics or emergency rooms.” As well, text that begins the second paragraph on page 7 states that “Overall, as we have said, the reviewed literature reflects mixed results . . . ” and that those inconclusive findings are a further reason for conducting our study. [Reviewer #2, comment #4]

  1. The revised paper supplies additional context in light of the journal’s international readership. For instance on page 5 (top), the text states that Medicaid and CHIP are designated for low-income families. It goes on to clarify that Medicaid and CHIP exclude undocumented immigrants from participation (page 7, middle of first paragraph). [Reviewer #2, comment #5]

  1. The revised version of the section describing our sample newly notes that the 2021 National Survey of Children’s Health is a public-use data set reflecting self-reported data provided by survey participants (see page 8, second paragraph). [Reviewer #2, comment #3]

  1. In our paper’s original data analysis section, it was explained clearly that we conducted a separate or independent multivariate logistic regression for each outcome variable (of which there were four). In the table accompanying the text, however, results obtained separately for each variable do appear together. For every regression we conducted, we also tested for multicollinearity of the explanatory variables in a model. These procedures determined that our analyses did not themselves raise the chance that a model or explanatory variable would demonstrate statistical significance. [Reviewer #2, comment #6]

  1. We revised the paper’s discussion section, striving for better explanation of our findings’ meaning and importance. For example, we now note (citing a study) that our findings suggest that “many immigrant children in the U.S. visited physicians or dentists but did not use prescribed medication. An explanation could be many such children’s probable use of traditional herbal or other home remedies instead of prescriptions from doctors”; see page 13, end of first paragraph. At the end of page 13’s second paragraph, our revised manuscript states “Such a result implies that immigrant children visit their dentists for regular check-ups.” Near the bottom of page 13, the revision states “These findings imply that immigrant parents attend to their young children’s health, while immigrant teenagers will act to maintain their own oral health and are willing to take medication prescribed by American doctors.” At the end of first paragraph on page 14, we now note that “…immigrant children would experience hospitalization when seriously sick or injured, regardless of their age.” [Reviewer #2, comment #7]

    After the first statement of second paragraph on page 14, text states “Close examination of the data revealed that the interaction term between Hispanic race/ethnicity and child dental condition (OR = 1.25, p < .05) was significantly associated (in positive direction) with likelihood of dentist visits. In other words, Hispanic immigrant children with poor oral health would not seek dental care.” [Reviewer #2, comment #7]

Our original discussion section has emphasized cultural beliefs upheld by many Asian immigrants: “One plausible explanation for such findings is Asian immigrants’ ethnicity-specific cultural beliefs, which often value traditional medicines and other traditional medical treatments over what the American medical system offers” (see page 14, third paragraph). In the same paragraph, we now have added an implication of our findings related to Black and other children of minority ethnicity in our sample of immigrants that they “elected to use hospital care in place of physician visits.” [Reviewer #2, comment #7]

Furthermore, near the top of page 15, our revision states that “The association indicates that the surveyed immigrant girls were as likely as the immigrant boys to use physician visits, hospital care, and prescribed medication, although not as likely to use dentist visits.” Similarly, at the end of the first paragraph on page 15, revised text reads “An implication of such findings is that relatively more-educated parents who have immigrated to the U.S. possess relatively strong knowledge or understanding of the health services available to their children through the American medical system.” At the end of the page’s second paragraph, moreover, we newly state that “Results of our study, then, imply that higher-income immigrant families can afford family health insurance. Additionally, they imply that public health insurance facilitates low-income immigrant children’s use of various health services.” [Reviewer #2, comment #7]

As a final example of our work to better emphasize our study’s meaning and importance, text on page 16 (first paragraph, middle) now explains that “…more-recently arrived immigrants are likely to elect hospital care for children rather than physician care.” [Reviewer #2, comment #7]

  1. Indications and implications of our findings (in discussion) are contributing to the policy and practice recommendations in conclusion section. The revised conclusion recommends making all documented immigrants eligible for Medicaid and CHIP, whatever the duration of their residence in the U.S. (see page 17, first paragraph, middle). Additionally, it recommends (a) that health-education programs offer classes focused on pertinent cultural beliefs and practices, citing a supportive published study; and (b) that awareness of children’s oral and physical health needs as well as awareness of the role of physician care be promoted among families recently arrived in the U.S. As well, at the end of the same paragraph, our conclusion now calls for the advocation and promotion of such family members’ enrollment in Medicaid, CHIP, and prescription-assistance programs. Moving on to next paragraph on page 17 (middle of the paragraph), the revised conclusion recommends (citing two published studies) accommodating use of traditional home remedies in conjunction with those treatments embraced by health services in the U.S. [Reviewer #2, comments #7]

  1. The revised manuscript is one page longer than the original, at 23 pages.

Reviewer 2 Report

This paper reports analyses of data on the use of healthcare services by children of immigrant families reported in the 2021 National Survey of Children’s Health in the USA.

The first thing to say about the paper is that it does not make it sufficiently clear in the title, or indeed anywhere else, that this study was done only in the USA, and therefore conclusions may be applicable only in that jurisdiction. It has been submitted to a journal that is titled as European. In my experience US-based researchers sometimes assume a hegemony by which their research naturally applies in other countries, and it often doesn’t. This is not malicious, but it is complacent and intellectually lazy. The title abstract methods and discussion sections need to reflect this reality.

Non-immigrant children are not included in these analyses, so the study makes no comparisons between these two groups. In this study possible explanatory variables are associated with measures of healthcare use exclusively within the population of immigrant children. I presume that reported use of services is being treated as a measure of access in the USA. In other jurisdictions, which have different immigration policies, and provide universal access to healthcare for immigrants, this can be interpreted differently (see below).  But the significance of higher or lower use of healthcare in the USA should be made clear in the introduction. Why exactly are they studying these associations? Is it to guide an expansion of publicly funded services?

As I understand it, all the data are based on survey (self-report) responses and (I presume because of anonymity) there is no linkage to routinely generated healthcare data. The authors should confirm this.  

The introduction to the study is a lengthy literature summary. I am not certain whether the authors confined their literature review to studies conducted in the USA, but details of their searching should be provided. The review is a rather soulless recounting of statistical associations between different variables and measures of healthcare use. I appreciate that these measures are being used, in part, to generate hypotheses for the present study but I think the introduction should be shorter and emphasize the importance and significance of these study findings.

As noted above, the report does not provide adequate context for an international audience. For instance, in several countries, e.g., Canada, Australia and (I assume) some European countries immigration applicants are subject to pre-screening and scoring according to particular skills and attributes. This does not apply to unauthorized immigrants and refugees. As a result, immigrants admitted under these systems have exhibited characteristics of ‘healthy cohorts’ and in some cases have reduced need for and use of healthcare services. In these settings reduced need does not denote reduced access. I believe that is not the case in the USA because of the immigration policies, which are a very vexed political topic. I think this should be discussed to establish both context and significance for an international audience.

The statistical methods used in the report seem sound and I have no critical comments. I will however point out that the authors explore many associations, so chance findings of statistical significance are likely. They have not applied an adjustment for multiple testing to test statistical significance.

The Discussion and Conclusions sections include some consideration of the importance and meanings of their findings but it’s quite short in comparison with other sections and lacks real depth. The policy implications are not discussed in adequate detail.

English language is fine but writing and presentation style makes the paper tedious to read

Author Response

(The authors gave the same response as above.)
